# Frequency Response of a Six-Electrode MET Sensor at Extremely Low Temperatures

**DOI:** 10.3390/s23094311

**Published:** 2023-04-27

**Authors:** Vadim Agafonov, Ivan Egorov, Anna Akinina

**Affiliations:** Moscow Institute of Physics and Technology, 141700 Dolgoprudny, Russia

**Keywords:** molecular-electronic technology, sensors, stabilization

## Abstract

Four-electrode electrochemical cells are widely used for signal conversion in molecular-electronic transfer (MET) motion sensors. The most used ACCA (anode–cathode–cathode–anode) configuration has proven its performance and usefulness for obtaining a superior conversion factor and a wider frequency range over standard geophones at room temperature. However, the MET sensor conversion factor decreases a thousand-fold or more when the temperature drops from room temperature to 233 K. In the design suggested is this paper, a pair of additional gate (G) electrodes has been added outside the standard ACCA cell. An experimental study of the temperature behavior of the resulting G-ACCA-G six-electrode configuration showed that the effects of temperature changes on the cell conversion factor are 5.2 times weaker compared with the standard ACCA configuration.

## 1. Introduction

Seismic sensors are used to measure small vibrations in applications such as seismology [1,2], mineral exploration [1,2,3], structural monitoring [4,5,6,7] and nuclear test monitoring. At the same time, these sensors are often used in field measurements, where they must have stable characteristics over a wide temperature range. In recent years, a type of seismic sensor known as an electrochemical sensor has become widespread. These sensors use liquid as the inertial mass and use electrochemical reactions at the electrodes as the physical mechanism for signal conversion [8]. As a result, the delivery of components to the electrodes depends on the motion of the liquid in the electrochemical cell that converts the signal, and consequently, on external vibrations. The advantages of electrochemical sensors are high sensitivity, including in the low-frequency region, simple and reliable mechanical design, and low requirements for the accuracy of the sensor installation with respect to the vertical.

In particular, at room temperature, the conversion coefficient of electrochemical sensors is 10–100 times [9,10,11,12] higher than that of an electromechanical geophone, which is the main instrument used in seismic exploration. At the same time, the sensitivity of electrochemical sensors is known to be determined by the viscosity of the working fluid and the diffusion coefficient of active ions involved in electrochemical electrode reactions [13]. The viscosity and diffusion coefficient depend on the temperature. The situation is different depending on the considered temperature range. If the operating temperatures are above 258 K, then an aqueous solution of potassium iodide is used as the working fluid, for which the viscosity and diffusion coefficients of ions change relatively little with temperature. Accordingly, electrochemical sensors retain their advantage in high sensitivity. For wider operating temperature ranges, LiI-based solutions and ionic liquids are used [14,15,16]. For them, the temperature dependences of the characteristics are more significant, and the sensitivity advantages of electrochemical sensors over electromechanical geophones are completely negated at low temperatures.

The problem of providing stable, temperature-independent characteristics is currently solved by using electronic circuits whose gain depends on temperature and frequency [17,18,19], thus compensating for the temperature changes in the properties of the working fluid. Additional stabilization of the characteristics is provided by using a force negative feedback [20,21,22]. The combined use of these approaches makes it possible to ensure the stability of the conversion coefficient down to 233 K, which, however, requires an increase in the gain in the electronics and inevitably leads to an increase in the noise of the sensor/electronics system.

In principle, the problem of decreasing sensitivity of electrochemical sensors and the related increase in noise can be solved by increasing the conversion coefficient of the primary sensitive element at low temperatures. In this paper, for these purposes, a six-electrode electrochemical cell has been proposed.

Compared with a standard four-electrode electrochemical cell, in the design studied in this paper, a pair of additional gate (G) electrodes has been added outside the standard ACCA (anode–cathode–cathode–anode) cell configuration.

The six-electrode configuration was first studied in [23]. The current–voltage characteristics (CVC) and amplitude–frequency response were measured for a six-electrode configuration at room temperature for a few different voltages between the gate electrodes and the anodes. Despite very limited experimental data, it was fundamentally shown that for such a six-electrode configuration, the sensitivity of the sensors can be changed by controlling the interelectrode potentials at the operating point. In the present work, the possibility of using a six-electrode cell configuration and a controlled change in the potentials on the electrodes is studied to improve the stability of the conversion coefficient in a wide temperature range from 296 K to 233 K.

An experimental study of the temperature behavior of the resulting G-ACCA-G six-electrode configuration showed that the temperature changes of the cell conversion factor are ~ five times weaker compared with the standard configuration. Consequently, sensors with a G-ACCA-G cell configuration have lower self-noise, with the difference especially significant at low temperatures.

## 2. Materials and Methods

The object of research is a six-electrode electrochemical cell (Figure 1). It consists of a standard four-electrode assembly (ACCA) and a pair of additional electrodes (G-G).

The four-electrode assembly consists of 2 cathodes (C-C) and 2 anodes (A-A), which are mesh platinum–iridium electrodes with a wire diameter of 45 μm and an area of 6 × 6 mm^2^. They are separated by 50 µm thick dielectric forsterite spacers to prevent short circuits. Forsterite gaskets contain through-holes with a diameter of 300 µm, which ensure the flow of electrolyte through the electrochemical sensing element. The electrolyte is a mixture of 0.4 mol/L of a highly concentrated aqueous solution of lithium iodide (LiI), the lower temperature limit of which is 218 K, and 0.1 mol/L of molecular iodine ($I_{2}$).

In a standard four-electrode unit, when a potential difference is applied between the anodes A and the cathodes C, an electrochemical current arises in the cell, the value of which depends on the concentration distribution of the active component in the near-cathode region.

Additional electrodes (G-G) are manufactured by applying conductive paste through stencils onto a fiberglass plate with holes for liquid flow. They are located on both sides of the four-electrode assembly, at a distance of 1 mm.

The six-electrode cell (G-ACCA-G) is fixed in a ceramic housing between membranes containing an electrolyte solution. It is shown in Figure 1.

In a six-electrode node, a change in the values of the potentials $U_{A}\;\mathrm{and}\;U_{G}$, on the pairs of anodes (A, G) leads to a change in the distribution of the active component near the anodes A. It was shown in [21] that in the electrochemical system under consideration, the ratio of the concentration of the active component of the solution on the electrodes A and G satisfies the condition:(1)$$\frac{C_{A}}{C_{G}}=\exp\left(\frac{2q_{e}}{kT}\left(U_{A}-U_{G}-\varphi_{A}+\varphi_{G}\right)\right),$$
where $\varphi_{A}$ and $\varphi_{G}$ are the electric potentials in the liquid solution at the boundary of the electrical double layer near the surface of the indicated electrodes.

Therefore, by changing the potential $U_{A}$, the concentration gradient at the cathodes and, thus, the interelectrode current and the sensitivity of the electrochemical cell can be controlled. Basically, this effect is similar to the dependence of the conversion coefficient on the concentration of the active component of the solution [10], with the difference that the average concentration remains unchanged, but can be locally changed at the electrodes by changing their potential.

In a standard four-electrode assembly, the choice of the potential $U_{A}$ is determined by the saturation current in the current–voltage curve. Similarly, to obtain the operating conditions of the six-electrode electrochemical cell and to determine the operating point, the current–voltage curves were studied. To do so, experimental stand No. 1 was used, which can be seen in Figure 2.

The electrodes of the electrochemical cell are connected to the measuring circuit, the signals from which were recorded by the NI USB-6215 analog-to-digital converter. The experiment was controlled by a computer program.

The measuring circuit consists of three electronic subcircuits:


*Block A*: converter of the current taken from the cathodes (C) into voltage, which is further digitized using ADC NI USB-6215 and recorded on a PC for further processing;*Block B*: converter of the current taken from the control anodes and anodes into voltage, which is further digitized using ADC NI USB-6215 and recorded on a PC to control the operation of the stand;*Block C*: a circuit that provides constant reference voltage at the anodes A using the built-in NI USB-6215 oscillator, and at the anodes G using resistors. When the four-electrode configuration is enabled, the anodes G are physically disabled.The potential $U_{A}$
was changed automatically using the built-in NI USB-6215 generator in the range of 40–380 mV. The operating mode settling time was 20 min, the data acquisition time was 20 min. The potential $U_{G}$ was changed manually using resistors.

To study the possibility of controlling the conversion factor of a six-electrode cell, the frequency response characteristics were measured at various potentials on the anodes A. To do so, the experimental stand No. 2, shown in Figure 3, was used. It contains a ‘magnet-coil’ system, which consists of a magnet attached to one of the membranes and an inductor coil, the current in which is set by an external electrical circuit and causes the electrolyte to flow through the electrochemical cell [24,25,26]. An alternating current of the required frequency was passed through the “magnet-coil” system in the range of 0.1–1000 Hz. The signal from the sensor was digitized using the ADC NI USB-6215, and then the frequency response characteristic was calculated automatically.

The measuring circuit consists of electronic subcircuits:*Block A*: current-to-voltage converter, which is further digitized using ADC NI USB-6215 and recorded on a PC for further processing. The amplitude of the differential cathode current is chosen as the output signal;*Block B*: converter of the current taken from the anodes A and the anodes G into voltage, which is further digitized using ADC NI USB-6215 No. 2 and recorded on a PC to control the operation of the stand;*Block C*: a circuit that provides the required potential value at the anodes A using the built-in NI USB-6215 generator No. 1, and at the anodes G using resistors. When the four-electrode configuration is enabled, the anodes G are physically disabled;*Block D*: a circuit that converts the output signal from the DAC (digital to analog converter) NI USB-6215 No. 2 into current, which is fed to the excitation coil [14,15,16], which interacts with a permanent magnet mounted on the membrane and provides the setting of the external motion on the electrochemical cell.

## 3. Results

First of all, a number of measurements were carried out to characterize the possibility of controlling the parameters of the transducer at room temperature. The current–voltage curves were measured according to the procedure described in the previous section for the following set of $U_{G}$ values: 80 mV, 120 mV, and 240 mV. The voltage $U_{A}$ varied in the range from 40 to 380 mV. The choice of this range is due to the fact that it is commonly used for studying the characteristics of a four-electrode assembly.

Thus, four CVCs were obtained at a temperature of 296 K (room temperature): for a four-electrode system and for a six-electrode system at $U_{G}=$ 80 mV, $U_{G}=$ 120 mV and $U_{G}=$ 240 mV. The measurement results are shown in Figure 4, where the value $U_{A}$ is indicated on the horizontal axis, while the sum of the currents from the cathodes C is indicated on the vertical axis at constant values of the potential $U_{G}$.

It is clear from Figure 4 that the behavior of the curves for different values of $U_{G}$ is similar to each other, but it is shifted along the horizontal axis by the value approximately equal to the difference of $U_{G}$ of the corresponding curves.

It is also clear from the graph in Figure 4 that for the four-electrode ACCA configuration, the saturation current is reached at 80 mV. Physically, the saturation current corresponds to the zero value of the stationary concentration of active ions on the cathodes and, at the same time, when the maximum value of the conversion coefficient is reached. As a consequence, typically, an operating point for the interelectrode voltage is chosen above the value at which the current reaches saturation. In practice, the potential value $U_{A}=300\;mV$ is most often used.

For the six-electrode G-ACCA-G configuration, the CVC behavior can be qualitatively explained using Equation (1). With $U_{A}<U_{G}$, the cathode current is less than its value obtained for a four-electrode system with the same $U_{A}.\;$This is due to the redistribution of the stationary concentration of active ions, namely, its decrease at the anodes and its increase at the gate electrodes according to Equation (1), which leads to a decrease in the diffusion flux of the active ions to the cathode. With equal potentials $U_{G}=U_{A},$ the CVCs for four-electrode and six-electrode systems intersect. At $U_{A}>U_{G}$, the cathode current in the six-electrode system exceeds the corresponding value for the four-electrode system due to the increase in the anode concentration according to Equation (1). Finally, at high values of $U_{A}$, the cathode current reaches its limiting value. In contrast to the four-electrode system, the occurrence of the limiting current cannot be associated with a decrease in the cathode concentration to zero since this happens at lower potentials when $U_{A}\sim \;80\;\mathrm{mV}$. This is due to an increase in the proportion of current flowing between the anodes and gate electrodes due to the increased natural convection in the anode space with a significant difference of $U_{A}-U_{G}$ and is related to these concentration variations, according to (1). Note that the limiting maximum currents for different curves related to the six-electrode system are close to each other and are approximately 1.7 times higher than the limiting current in the four-electrode system.

Given the similarity of the CVCs, we could expect that the ways to control the sensitivity of the converter by changing the anode potential at different potentials of the gate electrodes and the results of this control should be similar. Nevertheless, the use of $U_{G}=$120 mV gives some advantages compared with $U_{G}=80$ mV and $U_{G}=240$ mV, due to the wider range of $U_{A}$ available for variations to control the system parameters. This is because for $U_{G}=80$ mV, the range of $U_{A}<U_{G}$ does not ensure zero concentration conditions of the active ions on the cathodes, while at $U_{G}=240$ mV, the maximum cathode current could be achieved at voltages of $U_{A}$ higher than 400 mV. Using higher voltages may result in the initiation of a parasitic electrode reaction, which may lead to degradation of the system’s long-term parameters. Based on the above consideration, further studies were carried out at $U_{G}=$ 120 mV.

The measured frequency response at a temperature of 296 K (room temperature) for four-electrode (dashed line) and six-electrode (solid lines, different colors correspond to the changes in $U_{A}$ from 40 to 380 mV range) configurations are shown in Figure 5. As noted above, $U_{G}$ = 120 mV.

The presented graphs show that by controlling $U_{A},$ it is possible to increase the sensitivity of the transforming element by about four times. The effect was found to little depend on the frequency in the measurement range from 0.1 to 1000 Hz. A more detailed analysis at certain selected frequencies of 2 Hz, 0.2 Hz and 100 Hz showed, however, a somewhat stronger dependence at the frequency of 100 Hz on the control voltage, which is illustrated by the graph in Figure 6 and Table 1.

At the same time, a comparison of the behavior of the cathode background current (black curve) and conversion coefficients at different frequencies (red curves) shows that they correlate with each other. In the initial sections of the characteristic curve, an increase in $U_{A}$ noticeably increases the background cathode current and conversion coefficients. Starting approximately from $U_{A}=180\;\mathrm{mV}$, all the studied quantities reach a relatively stationary level.

Further measurements were carried out in the temperature range from 296 K to 233 K with a step of 10 degrees. To set and maintain the required temperatures, a climatic chamber M-60/100-120 KTX-T, manufactured by Mir Oborudovania LLC, Sankt-Petersburg, was used. The temperature maintenance accuracy was ±1 °C.

The obtained CVCs are presented in Figure 7, where the voltage at the anodes A is indicated along the horizontal axis, and the total current from the cathodes C in mA on a logarithmic scale is indicated along the vertical axis. The dashed lines indicate the limiting currents for the four-electrode system at the corresponding temperatures (temp).

Qualitatively, the behavior of the current–voltage characteristics is consistent with the results obtained for room temperature. With a decrease in temperature at given operating voltages, the currents quite expectedly decrease, which is associated with a decrease in the diffusion coefficient. The graphs presented also show that temperature changes in the cathode current can be partially compensated if the operating point on the curves is shifted to the right with decreasing temperature, i.e., by increasing the potential of $U_{A}$. As a particular example, it appears from the data obtained that it is possible to fully compensate for the change in the background current when the temperature drops from 296 K to 263 K, if, simultaneously with the temperature decrease, the potential $U_{A}$ is increased from 40 to 380 mV. In a wider temperature range, it is possible to partially compensate for changes in the cathode current. Taking into account the above-mentioned correlation between the background cathode currents and the conversion factor (discussed in relation to the results shown in Figure 7), this approach should not only reduce the change in the background current, but also weaken the temperature dependence of the conversion factor.

To determine the change in the coefficient under the condition of a temperature-controlled change in $U_{A}$, the frequency response was measured at $U_{A}=40\;mV$ for room temperature and at $U_{A}=320\;mV$ for −40 °C. The resulting dependences of the frequency response are shown in Figure 8 (red and blue curves, respectively).

At the same time, the frequency responses for a four-electrode system for the same temperatures are presented. As a result, for a frequency of 2 Hz, for example, the conversion factor changes by a factor of 2400 for a standard four-electrode system, while the conversion factor for a six-electrode system changes by a factor of 460.

## 4. Conclusions

Thus, for this node configuration with an electrolyte consisting of 4 mol/L of a highly concentrated aqueous solution of lithium iodide (LiI) and 0.1 mol/L of molecular iodine ($I_{2}$), we can state that by controlling the potential at the anodes A at a fixed potential at the electrodes G, the concentration of the active component on the anodes can be changed, which enables control of the value of the cathode current and the conversion coefficient. It was experimentally confirmed that by applying the temperature-dependent law of change in the control potential, the change in the conversion coefficient of electrochemical cell motion sensors with temperature change can be significantly changed, up to five times. Note that a more stable behavior of the conversion factor is achieved without changing the conversion factor of the accompanying electronics and the associated effect of the self-noise of the electronics on the recorded signal.

Based on the results obtained here, there is, apparently, no fundamentally achievable limit to the stabilization of the conversion coefficient with temperature. From the physical point of view, the stationary value of the anode concentration is the result of establishing a balance between the creation of active ions. This results from the potential difference between the anodes and additional electrodes stimulated by an electrochemical reaction and the escape of ions from the region near the anodes due to diffusion and convective processes. It is likely that the concentration growth effect can be significantly enhanced with the optimal location of the electrodes in the electrochemical conversion cell. Studying this possibility should be the aim of further research.

## Figures and Tables

**Figure 1 sensors-23-04311-f001:**
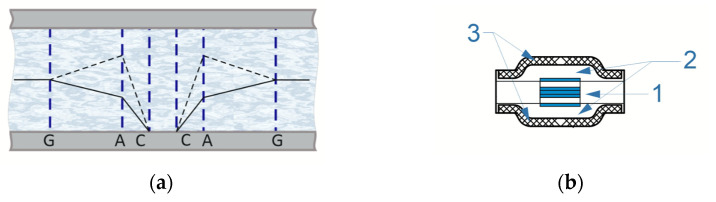
(**a**) Six-electrode electrochemical cell. (**b**) The electrochemical cell design: 1—six-electrode electrochemical cell enclosed in a ceramic case, 2—electrolyte solution, 3—membranes.

**Figure 2 sensors-23-04311-f002:**
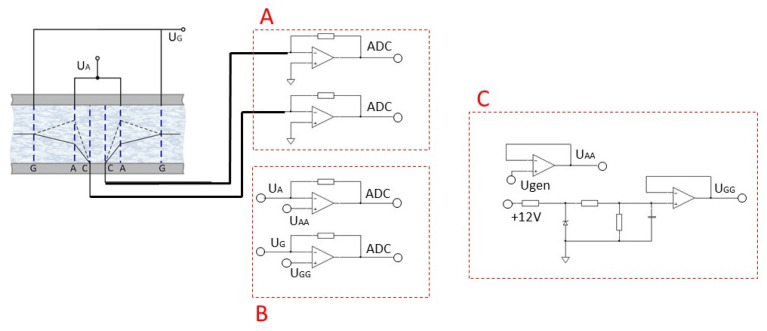
Schematic diagram of the experimental stand No. 1, which was used to measure the CVC of a six-electrode cell. Block A—converter of the current taken from the cathodes (C) into voltage; Block B—converter of the current taken from the control anodes and anodes into voltage; Block C—a circuit that provides a constant reference voltage at the anodes (A).

**Figure 3 sensors-23-04311-f003:**
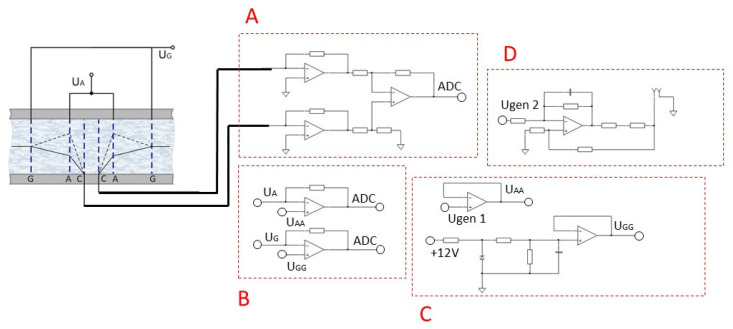
Schematic diagram of the experimental stand No. 2, which was used to measure the frequency response of a six-electrode cell. Block A—current-to-voltage converter; Block B—converter of the current taken from the anodes A and the anodes G into voltage; Block C—a circuit that provides the required potential value at the anodes (A); Block D—a circuit that converts the output signal from the digital to analog converter into current.

**Figure 4 sensors-23-04311-f004:**
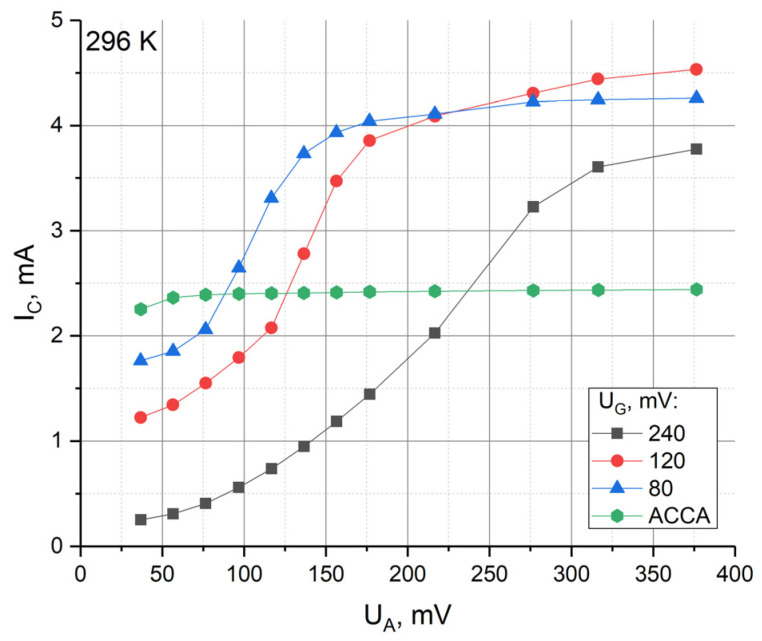
CVC for 6- and 4-electrode systems.

**Figure 5 sensors-23-04311-f005:**
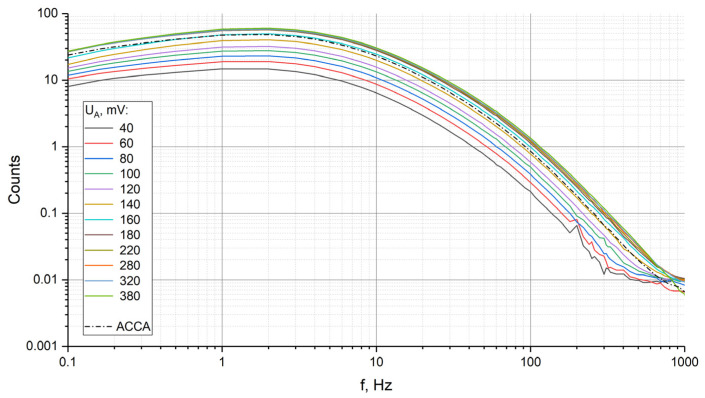
Frequency response family for 6- and 4-electrode configurations at 296 K.

**Figure 6 sensors-23-04311-f006:**
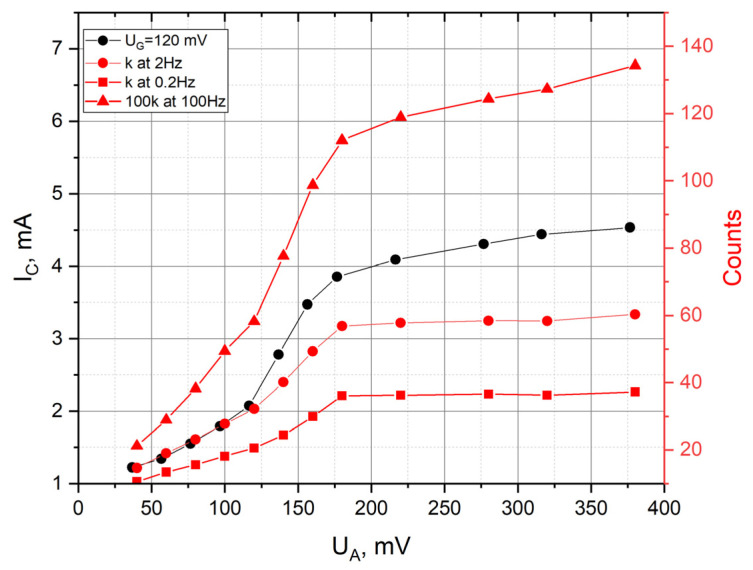
CVC of a 6-electrode cell, on the anodes G 120 mV and the conversion factor k at 2 Hz, 0.2 Hz and 100 Hz.

**Figure 7 sensors-23-04311-f007:**
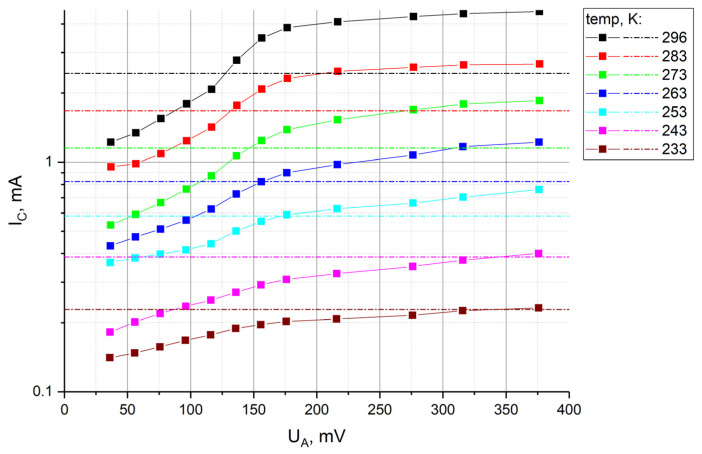
CVCs of a 6-electrode cell under various temperature conditions.

**Figure 8 sensors-23-04311-f008:**
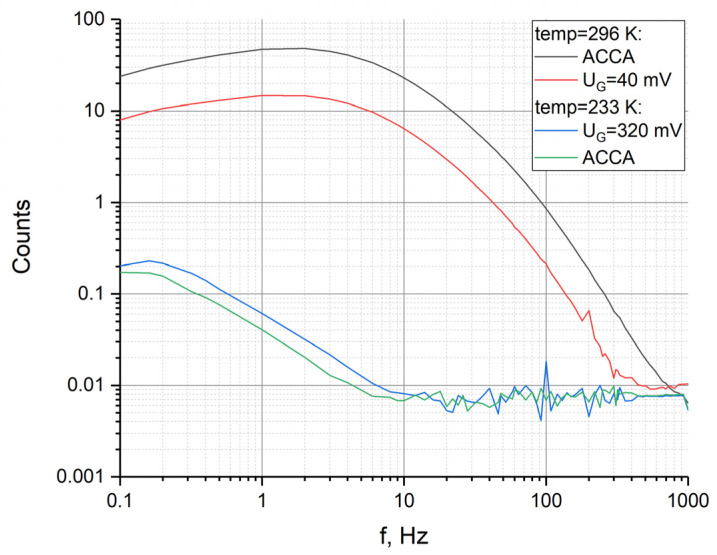
Frequency responses for 4- and 6-electrode assembly at temperatures of 233 K and 296 K.

**Table 1 sensors-23-04311-t001:** The conversion factor k at 2 Hz, 0.2 Hz and 100 Hz.

Hz	$\frac{k\;\left(320\;mV\right)}{k\;\left(40\;mV\right)}$
0.2	3.98
2	3.41
100	6.01

## Data Availability

Not applicable.

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
