# Peer review of "Frequency Response of a Six-Electrode MET Sensor at Extremely Low Temperatures"

_sensors, 2023, doi:10.3390/s23094311_

Round 1

Reviewer 1 Report

This paper presents development of temperature independent (weaker dependency) MET sensor response when switching from standard 4-electrode ACCA (anode-cathode-cathode-anode) configuration to 6-electrode configuration with the introduction of a pair of additional gate electrode outside ACCA cell to achieve G-ACCA-G configuration. The scientific aspect of the paper looks good, but the presentation makes it difficult to follow. This paper could be accepted after following modifications.

1)  The figure legends are not clear, and the axis of the figure (4 to 8) is difficult to follow.

2)  Author should describe the reason for selecting UG=120mV, the given explanation is not sufficient.

3)   Kindly rephrase the sentence from page 5 “the behaviour of the curves for different values of UG is similar to each other” to represent the negligible variation in the trend with its explanation.

4)    The full form of the abbreviation CVC is not mentioned in the manuscript.

Author Response

â„–

Reviewer comments

Answers and Resulted modification

1)

The figure legends are not clear, and the axis of the figure (4 to 8) is difficult to follow.

Authors thank the reviewer for a valuable comment. We have added a more detailed and clear legend of the figure (4 to 8).

2)

Author should describe the reason for selecting UG=120mV, the given explanation is not sufficient.

Authors thank the reviewer for the suggestion. We have added a detailed explanation of the criteria to select the operating voltages. Generally, the approach is to use UG, which gives a wider range for changes of UA. The details are presented in the text after Figure 4 (lines 183 to 218).

3)

Kindly rephrase the sentence from page 5 “the behaviour of the curves for different values of UG is similar to each other” to represent the negligible variation in the trend with its explanation.

Authors thank the reviewer for a valuable comment. We have added a more detailed explanation related to the curves behavior. The following text has been added immediately after Figure 4 (lines 177 to 179).

“It is clear from Figure 4 that the behavior of the curves for different values of  is similar to each other, but it is shifted along the horizontal axis by the value approximately equal to the difference of  related to the corresponding curves.”

4)

The full form of the abbreviation CVC is not mentioned in the manuscript.

Authors thank the reviewer for the suggestion. We have added the full form of the abbreviation CVC: Current–voltage characteristic (line 61).

Reviewer 2 Report

This paper presents a new design of a six-electrode MET sensor and compares its performance with a four-electrode sensor at low temperatures. The operating conditions of the six-electrode sensor is determined, and its sensitivity can be adjusted by changing the potential. The paper is basically well-written and organized. But there are some issues need to be addressed before the acceptance:

1. The first occurrence of abbreviations in the article needs to be explained, such as MET and CVC.

2. In the "Abstract" section, 'several times' is better to be specified with some exact numbers.

3. Adding titles for the horizontal and vertical axes in the figures and tables would be necessary.

4. The English of the manuscript need to be greatly improved.

5. The usage of “up to 233 K” in line 51 may mislead readers that the stability of the conversion coefficient is a great challenge at high-temperature environment.

6. Line 61, reference [23] has reported the 6-electrode configuration. What’s the contribution of this work compared with the reference [23]?

7. It’s better to mark the diagram in figure 1 into figure 1(a) and (b) to make the manuscript more readable.

8. It’s better to have a more detail caption for figure 2 and 3.

9. The layout of Figure 6 and Table 1 is incorrect, and there are some typos in the Table 1.

10. A section of “Conclusions” should be included in the manuscript.

Author Response

â„–

Reviewer comments

Answers and Resulted modification

1)

The first occurrence of abbreviations in the article needs to be explained, such as MET and CVC.

Authors thank the reviewer for the suggestion. We have added the full form of the abbreviation MET: molecular- electronic transfer (line 8) and CVC: Current–voltage characteristic (line 61).

2)

In the "Abstract" section, 'several times' is better to be specified with some exact numbers.

Authors thank the reviewer for a valuable comment. We have added more detailed numbers (line 16).

3)

Adding titles for the horizontal and vertical axes in the figures and tables would be necessary.

Authors thank the reviewer for a valuable comment. We have added the titles for axes in the figures and tables.

4)

The English of the manuscript need to be greatly improved.

Authors thank the reviewer for a valuable comment. The certified translator checked and edited the manuscript.

5)

The usage of “up to 233 K” in line 51 may mislead readers that the stability of the conversion coefficient is a great challenge at high-temperature environment.

Authors thank the reviewer for a valuable comment. We have changed the usage of  “down to 233 K” in line 51.

6)

Line 61, reference [23] has reported the 6-electrode configuration. What’s the contribution of this work compared with the reference [23]?

Authors thank the reviewer for a valuable comment. We have added the contribution of this work compared with the reference [23] (line 61).

7)

It’s better to mark the diagram in figure 1 into figure 1(a) and (b) to make the manuscript more readable.

Authors thank the reviewer for a valuable comment. We have marked the diagram in Figure 1 into Figure 1(a) and (b) to make the manuscript more readable.

8)

It’s better to have a more detail caption for figure 2 and 3.

Authors thank the reviewer for a valuable comment. We have added a more detail caption for Figures 2 and 3.

9)

The layout of Figure 6 and Table 1 is incorrect, and there are some typos in the Table 1.

Authors thank the reviewer for the suggestion. We have corrected the layout of Figure 6 and Table 1.

10)

A section of “Conclusions” should be included in the manuscript.

Authors thank the reviewer for a valuable comment. We have added a section of “Conclusions” in the manuscript.

Round 2

Reviewer 1 Report

The authors addressed the comments and the paper is looking good for publication after a small modification to Figures 5 and 6 as they are repeated twice.

Author Response

Authors thank the reviewer for a valuable comment.  We checked the text of the article and uploaded the correct version of the file, where there is no duplication of figures.
